# Distributed Optimization in Low Voltage Distribution Networks via Broadcast Signals †

**Boyuan Wei ‡ and Geert Deconinck \*,‡** 

Department of Electrical Engineering (ESAT), Research division Electa, KU Leuven, 3001 Leuven, Belgium;
Boyuan.wei@kuleuven.be

\* Correspondence: geert.deconinck@kuleuven.be
† This paper is an extended version of our paper published in CIGRE Chengdu Symposium 2019.
‡ Kasteelpark Arenberg 10, bus 2445, 3001 Leuven, Heverlee, Belgium.

**Abstract:** With the development of distributed energy resources, the low voltage distribution network (LVDN) is supposed to be the integrator of small distributed energy sources. This makes the users in LVDNs multifarious, which leads to more complex modeling. Additionally, data acquisition could be tricky due to rising privacy concerns. These impose severe demands on control schemes in LVDNs that the classical centralized control might not be able to fulfill. To tackle this, a model-free control approach with distributed decision-making architecture is proposed in this paper. Employing statistical methods and game theory, individual users in LVDNs achieve local optimum autonomously. Comparing to conventional approaches applied in LVDNs, the proposed approach is able to achieve active control with less communication burden and computational resources. The paper proves the convergence to the Nash Equilibrium (NE) and uses player compatible relations to form the specific equilibrium. A variant of the log-linear trial and error learning process is applied in a novel "suggest-convince" mechanism to implement the proposed approach. In the case study, a 103 nodes test network based on a real Belgian semiurban LVDN is illustrated. The proposed approach is validated and analyzed with practical load profiles on the 103 nodes network. In addition to that, centralized control is implemented as a benchmark to show the performance of the proposed approach by comparing it with the classical optimization result. The results demonstrate that the proposed approach is able to achieve player compatible equilibrium in an expected way, resulting in a good approximation to the local optimum.

**Keywords:** low voltage distribution network; player compatible equilibrium; voltage control; active distribution network; Nash equilibrium; broadcast signals

## 1. Introduction

The rising penetration level of distributed energy resources (DER) has become a clear trend in modern distribution networks. This has been especially challenging controlling low voltage distribution networks (LVDNs) as LVDNs are expected to facilitate the penetration of DER. On one hand, higher DER penetration levels in LVDNs are synonymous with greater variability, which need larger flexibility reserve and active control. On the other hand, consumer privacy becomes an important concern when controlling users with the deployment and adoption of smart grid technologies, which set severe barriers to data collection from users [1]. User uncertainty and the awareness of privacy bring more difficulties in modeling and information acquisition, both of which are critical in conventional active control approaches [2]. Thus, there is a demand for active control schemes with limited information or even without specific models (model-free).

There are quite a few existing approaches that are deemed to be able to fulfill such a demand at certain degrees. Droop control [3,4] is a classic and reliable way to implement distributed control without relying on communication. Nevertheless, although drawbacks like load sharing can be solved by some modified scheme [5,6], some issues remain. For instance, most of DERs, if not all, have no droop character originally, and the control performance is affected by cable impedance [7]. Moreover, as a passive approach, some notable features of the smart grid such as active management and dynamic optimization are not easy to implement by droop control. Most of the existing active control is centralized and comes with high communication demand, aiming at offering ancillary services [8,9]. Additionally, the modeling of users is a remaining concern especially when they are multifarious.

All of these limits and restrictions suggest that attempts to achieving distributed model-free control is needed. As the decision-making process of control is decentralized, information is collected and used locally in a distributed manner. This helps the predicament on both communication burden and privacy concerns. Furthermore, with the ability to perform parallel computations, distributed algorithms have the potential to be computationally superior to centralized algorithms, both in terms of solution speed and the maximum problem size that can be addressed [10]. Distributed control, including optimization, has found its applications in power systems especially on electrical vehicle charging and demand response. Readers are referred to the survey literature [11,12] for more instances. Most of these works are based on computational models or techniques, such as Augmented Lagrangian Decomposition [13] and the decentralized solution of the Karush–Kuhn–Tucker (KKT) necessary conditions for local optimality [14]. Besides these computational approaches, as a study of strategic interaction among rational decision-makers, game theory finds its application in distributed control structure firmly. Ref. [15] proposed a strategy on distributed energy allocation between providers and consumers, while demand response among residential consumers considering irrational behavior is studied in [16], and a set of distributed robust adaptive computation algorithms for a class of generalized convex games by computing the Nash Equilibrium is proposed by [17]. In this paper, a control scheme under distributed decision-making architecture is studied and the Nash Equilibrium (NE) is involved to drive users toward local optimality on flexibility management in a given LVDN.

As one of, if not the most, famous studies in game theory, NE has been attracting research interests for several decades in various fields. Generally, there are two major threads on seeking NE in practical applications: employing mathematical framework and solving a class of generalized convex games locally or globally, which is so-called "mathematical approaches" and designing rules of learning or evolving that can strike dynamic equilibrium within finite iterations, which sometimes is described as "trial and error learning". The "mathematical approaches" are widely studied in the control field. For instance, the ODE-based generalized NE computation [18], nonlinear Gauss–Seidel-type approach [19], best-response dynamics [20], generalized convex games over unreliable networks [17], and using Newton-type methods to find NE at a super-linear convergence rate [21]. The "trial and error learning" have myriad applications in economics and has been employed in engineering fields over time. Hart, Foster, and Young's research [22–24] have proved decentralized rules can be devised that converge to NE or correlated equilibrium in general n-person games. [25] uses the classic model of rational Bayesian to maximize the discounted expected utility under the belief that the environment is constant. Based on NE, Player-Compatible Equilibrium (PCE) was proposed by [26], which extends the consideration of "trembles" in the NE by imposing cross-player restrictions on the game, in a way that is invariant to the utility representations of players' preferences over game outcomes. A trembling hand perfect equilibrium is an equilibrium that takes the possibility of off-the-equilibrium play into account by assuming that the players, through a "tremble", may choose unintended strategies, albeit with negligible probability. Readers are referred to [27] for more explanations.

Although there are quite a few existing works that involve game theory in power systems, most of them focus on electricity market issues [28–32] or employ game theory as auxiliary to the main control algorithm [33]. For instance, Ref. [34] employs game theory in a hybrid energy system for voltage and frequency control, while the game theory algorithm has only been used to decide which

energy source should be used at a certain point. Meanwhile, most of the works use one of Shapley [32], Aumann–Shapley [35], or Nucleolus-based algorithms [36].

This paper attempts to use novel game theoretic algorithms to tackle technical, non-economic issues in LVDNs. The contribution of this paper is three-fold. Firstly, a control scheme that can be implemented by LVDN users in a distributed manner, under a broadcaster-users architecture is proposed. To make it clear, the concept "users" in this paper indicates all the households and individual devices connected to LVDN, including small distributed generators, PV, small wind turbines, and so on. Without massive communication and complicated modeling, users employ local information and simple public information broadcast by the broadcaster to decide their own strategies independently. As no specific information is required in the control scheme, hot-plugging is feasible, which allows users to join/quit the network freely, making LVDNs flexible. Secondly, the paper proves that the proposed scheme is able to drive users to converge to PCE within the limited period to achieve the specific control objectives. Thirdly, a benchmark with centralized optimization is presented in this paper, with remarks on the performance comparison. The paper is organized as follows: the specific problem is elaborated in Section 2; Section 3 introduces the necessary concepts and then illustrates the scheme of the proposed approach; a simulation study based on a practical network and benchmark are presented in Section 4, and then Section 5 concludes the paper.

## 2. Problem Statement

### 2.1. Notations

Assume a strategic-form game with $N$ players, and each player $i$ has a finite strategy set $\mathbb{S}_i$, where strategy $s_i \in \mathbb{S}_i$. The set of mixed strategy $\sigma_i$ is $M_i$ and the set of strictly mixed strategies, where every pure strategy in $\mathbb{S}_i$ has nonzero probability, is denoted by $M_i^\circ$. For $s_i$, the correlated strategies of the other $N-1$ users is $s_{-i}$. The cost of user $i$ is indicated by $J_i$, the utility of user $i$ is $U_i = -J_i$.

### 2.2. Problem Statement

Consider an LVDN with N users, where each user is located in a different location. The users are not necessarily homogeneous but consume or provide power on a comparable order of magnitude. Each user $i$ has a power consumption profile $\tilde{p}_t^i$, where generation is regarded as negative consumption in this paper. Assume the given LVDN has limited flexibility reservation. It means if all the users operate as they want, there will be a mismatch in the balance, which leads to voltage issues. In LVDN, due to the high R/X ratio, voltage is more sensitive to active power balance, but still related to reactive power as well [37,38]. A distributed control is needed to optimize the operations among users to strike a balance between user comfort and voltage regulation. Poor user comfort leads to dissatisfaction. Users will become less satisfied if they cannot operate as they wish, and partial operation or stepless adjustment of power is not feasible. For instance, if user $i$ wants to increase its power from 1.5 to 2 kW, increasing its power to 1.8 kW instead will not make it satisfied and there is no guarantee that 1.8 kW is a feasible working state for user $i$. This is very common for most of the household appliances or distributed generators. Meanwhile, the communication and computation should not be too intensive for easy implementation.

Essentially, power regulation in LVDNs can be abstracted as a flexibility allocation game. If user $i$ changes its working status against the power balance, it consumes flexibility. If the flexibility reserve is not sufficient, this leads to a voltage problem. Conventionally, the LVDN is supported by a backbone distribution network, which is supposed to provide sufficient flexibility. Whereas the flexibility allocation can be optimized to achieve a more robust and independent LVDN, which is in line with the concept of the active distribution network (ADN) [39]. In this game, each user $i$ has to implement its strategy $s_i \in \mathbb{S}_i$ in every control cycle, which is decided by a centralized optimization algorithm in

conventional control schemes. According to $\tilde{p}_t^i$, each user $i$ has an initial plan $\delta_t^i$, which denotes the original profile change tendency of user $i$. In the given problem of this paper, $\mathbb{S}_i$ is a finite set, then

$$p_{t+\Delta t}^i = \begin{cases} p_t^i + \Delta p^i & , s_i = 1 \\ p_t^i & , s_i = 0 \\ p_t^i + \delta_t^i & , s_i = -1 \end{cases} \quad , \text{where } \delta_t^i = \tilde{p}_{t+1}^i - \tilde{p}_t^i. \tag{1}$$

$p_t^i$ is the actual power consumption of user $i$, where $t + \Delta t \in (t, t+1)$. $\Delta p^i$ is the admissible regulation that proposed by control. Thus, the cost of user $i$ is given by

$$J_i(p_{t+\Delta t}^i, v_{t+\Delta t}^i)dt = a \cdot (p_{t+\Delta t}^i - \tilde{p}_{t+1}^i)^2 \cdot dt + b \cdot (v_{t+\Delta t}^i - \xi)^2 dt, \ \xi = (1 + sgn(v_t^i - v^r) \cdot 0.1\gamma) \cdot v^r, \tag{2}$$

where $v_{t+\Delta t}^i$ is the corresponding local nodal voltage of $i$ after $s_i$ is implemented, and $0 < 2a < b$. $\gamma \in [0, 0.99]$ is the boundary coefficient. The objective of each user $i$ is to minimize its $J_i$. Although $p_{t+\Delta t}^i$ is controlled by $s_i$, user $i$ cannot always choose $s_i = 1$ to minimize $J_i$, as $b < 2a < 0$ and if $v_{t+\Delta t}^i$ turns out being far away from $\xi$, $J_i$ grows.

## 3. Concepts and Control Scheme

### 3.1. Concepts Preparation

**Proposition 1.** *Resource allocation game [40] is a congestion game.*

Congestion games are a general model for resource allocation games and are a special class of potential games [41].

**Theorem 1.** *A potential game converges to a Nash Equilibrium (NE). Games that are close (in terms of payoffs of players) to potential games have similar limiting dynamics to those in potential games.*

For simplicity, readers are referred to the Theorem 3.1 in [42] for definitions and corresponding proofs.

**Definition 1.** *For user $i \neq j$ and the corresponding strategies $s_i^* \in \mathbb{S}_i, s_j^* \in \mathbb{S}_j$, if it holds for every correlated strategy $\sigma_{-j} \in M_{-j}^\circ$ and for every $\sigma_{-i} \in M^\circ(\mathbb{S}_{-i})$ satisfying $s_{-i}|\mathbb{S}_{-ij} = s_{-j}|\mathbb{S}_{-ij}$, that*

$$U_j\left(s_j^*, \sigma_{-j}\right) \geq \max_{s_j' \in \mathbb{S}_j \setminus \{s_j^*\}} U_j\left(s_j', \sigma_{-j}\right), \text{ while } U_i(s_i^*, \sigma_{-i}) > \max_{s_i' \in \mathbb{S}_i \setminus \{s_i^*\}} U_i(s_i', \sigma_{-i}), \tag{3}$$

then we say $i$ is more *player compatible* with $s_i^*$ than $j$ is with $s_j^*$, which is denoted as $(s_i^*|i) \succsim (s_j^*|j)$. This compatibility relation is transitive and asymmetric, as the following propositions. Readers are referred to the appendix in [26] for the corresponding proofs.

**Proposition 2.** *If $(s_i^*|i) \succsim (s_j^*|j) \succsim (s_l^*|l)$, then $(s_i^*|i) \succsim (s_l^*|l)$.*

**Proposition 3.** *If $(s_i^*|i) \succsim (s_j^*|j)$, then $(s_j^*|j) \not\succsim (s_i^*|i)$.*

**Proposition 4.** *If the game only has two players $i \neq j$, then $(s_i^*|i) \succsim (s_j^*|j) \succsim (s_l^*|l)$ never holds, as this concept considers third parties, whose best response is affected by the relative tremble probabilities of $i \neq j$.*

**Definition 2.** *Assume there is a tremble profile $\epsilon$, which assigns a positive number (no matter how small it is) $\epsilon(s_i|i) > 0$ to $s_i$ of every player i, use $\Pi_i^\epsilon$ for the set of these strategies of player i, then we write*

$$\Pi_i^\epsilon := \{\sigma_i \in M(\mathbb{S}_i) \ s.t. \ \sigma_i(s_i) \geq \epsilon(s_i|i) \ \forall s_i \in \mathbb{S}_i\}. \tag{4}$$

*Then $\sigma^\circ$ is $\epsilon$-equilibrium, if*

$$\sigma_i^\circ \in \arg\max_{\sigma_i \in \Pi_i^\epsilon} U_i(\sigma_i, \sigma_{-i}^\circ) \tag{5}$$

*$\Pi_i^\epsilon$ is convex and compact. Whenever $\epsilon$ is small enough so that $\Pi_i^\epsilon$ is non-empty for every i, the existence of $\epsilon$-equilibrium holds.*

**Definition 3.** *$\epsilon$ is player compatible if $\epsilon(s_i^*|i) \leq \epsilon(s_j^*|j)$ for all $i, j$, and $s_i^*, s_j^*$ such that $(s_i^*|i) \succsim (s_j^*|j)$. Then a $\epsilon$-equilibrium where $\epsilon$ is player compatible is called player compatible $\epsilon$-equilibrium ($\epsilon$-PCE).*

**Theorem 2.** *PCE exists in every finite strategic-form game [26].*

**Definition 4.** *State $z(t) \in Z^*$ is stochastically stable state if for every i and given any small $\varphi > 0$,*

$$U_i(s_i^*, \sigma_{-i}) \geq \max_{s_i' \in \mathbb{S}_i \backslash \{s_i^*\}} U_i(s_i', \sigma_{-i}) \tag{6}$$

*holds for at least the fraction $1 - \varphi$ of all periods $\tau$.*

*3.2. Control Scheme*

3.2.1. Architecture Setup

Besides the users in LVDN, we assume that there is a global broadcaster who periodically monitors the voltages $v_t^k$ from $K$ key points over the network at time $t$, then calculates the general situation parameter $g_t$ according to

$$g_t = (1 - sin(\frac{e^{-(\frac{\mu}{10})^2}}{2}\pi)) \cdot sgn(\mu), \ where \ \mu = \overline{v}_t - v^r. \tag{7}$$

$\overline{v}_t$ is the average of $v_t^k$, $v^r$ is the rated voltage so that $g_t$ suggests the general voltage situation of whole LVDN. Then $g_t$ is broadcast to every user as public information, users decide and implement their own strategies independently within $\mathbb{S}_i$. There is no extra communication besides the broadcast needed.

3.2.2. Control Scheme

As there is no communication besides broadcast, the decision-making process of users is a non-cooperative game. The public information provided by broadcast promotes coherence among users, as the decision-making process is decentralized compared to centralized control approaches, whose coherence is guaranteed by centralized decision making and bi-directional communication.

As stated in the previous section, the control can be regarded as a flexibility allocation game. According to Proposition 1, this turns out to be a congestion game, which is a special class of potential game. Theorem 1 suggests that a potential game converges to a NE. NE is a stable state of a system involving the interaction of different participants, in which no participant can gain by a unilateral change of strategy if the strategies of the others remain unchanged. According to the definition of NE, it is reasonable to conclude that $J_i$ has achieved its minimum if the game converges to a NE. However, just like local optimum in numerical optimization, there might be multiple NE in one potential game. Therefore, the proposed control scheme has to fulfill two requirements: firstly, the game should be able to converge to NE within an acceptable time, secondly, there should be a guarantee that the game converges to a specific admissible NE.

One can figure out that the minimum of $J_i$ is obtained, where $s_i = -1$ and $v^i_{t+\Delta t} \longrightarrow \xi$. In other words, user $i$ reaches its minimum cost when user $i$ follows its own plan and the local voltage is close to the set point. Practically speaking, this is the idea of user deregulation. In this paper a so-called "suggest-convince" mechanism is proposed to configure the decision making of user $i$, to minimize $\int_0^\tau J_i(p^i_{t+\Delta t}, v^i_{t+\Delta t})dt$ in (2), where $\tau$ is the period.

$g_t$ is the incentive in the proposed control. Whenever user $i$ receives $g_t$, it will firstly operate Algorithm 1 independently to figure out the suggestion parameter $\zeta^i_t$, in which Equation (8) is given by

$$\Delta v^i_t = (1 - sin(\frac{e^{-(\frac{\mu}{10})^2}}{2}\pi)) \cdot sgn(\mu), \ where \ \mu = v^i_t - v^r. \tag{8}$$

Step 5 in Algorithm 1 works in a seesaw manner. When $v^i_t$ has a large deviation from rated voltage, it is more dominating in the computation of $\zeta^i_t$, otherwise, $g_t$ will be dominating. This suggests that user $i$ always regards its own situation as a priority.

---

**Algorithm 1** Suggest Phase

---

**Input:**
    Public voltage information, $g_t$;
    Local voltage, $v^i_t$;
    Rated voltage, $v^r$;
**Output:**
    Suggestion parameter, $\zeta^i_t$;
 1: calculate $\Delta v^i_t$ according to equation (8)
 2: **if** $|\Delta v^i_t| > 0.99$ **then**
 3:     $\Delta v^i_t = sgn(\Delta v^i_t)$;
 4: **end if**
 5: $\zeta^i_t = (1 - |\Delta v^i_t|)g_t + |\Delta v^i_t| \cdot \Delta v^i_t$;
 6: Return $\zeta^i_t$

---

Each user $i$ maintains a two dimensional vector $\Lambda_i = \left( \begin{bmatrix} \lambda^i_{11} \\ \lambda^i_{21} \end{bmatrix}, \begin{bmatrix} \lambda^i_{12} \\ \lambda^i_{22} \end{bmatrix}, \cdots [\lambda^i_m] \right)$, where $m \in [1, 2, 3 \cdots K], [\lambda^i_m] = \begin{bmatrix} \lambda^i_{1m} \\ \lambda^i_{2m} \end{bmatrix}, \lambda^i_m \in (0,1), \lambda^i_{1m} > \lambda^i_{2m}$. This is so called stubborn vector. The suggestion parameter $\zeta^i_t$ that is figured out by Algorithm 1, which indicates the suggested adjustment, is used to "convince" the corresponding stubborn vectors according to Algorithm 2.

---

**Algorithm 2** Convince Phase

---

**Input:**
    Suggestion parameter, $\zeta^i_t$;
    Stubborn vector, $\Lambda_i$;
    Period index, $m$;
**Output:**
    Strategy, $s_i$;
 1: **if** $\zeta^i_t > \lambda^i_{1m}$ **then**
 2:     $s_i = 1$;
 3: **else if** $\lambda^i_{1m} \geq \zeta^i_t > \lambda^i_{2m}$ **then**
 4:     $s_i = 0$;
 5: **else if** $\lambda^i_{2m} > \zeta^i_t$ **then**
 6:     $s_i = -1$;
 7: **end if**
 8: Return $s_i$

---

Depending on $s_i$, the corresponding control will be implemented according to (1). Nevertheless, $\Delta p^i$ needs to be specified if $s_i = 1$, as $\zeta_t^i$ only indicates the direction. The specific $\Delta p^i$ is figured out in Algorithm 3. One can find that it is possible for $\Delta p^i = \delta_t^i$ even when $s_i = 1$, while it is different from the situation when $s_i = -1$, this will be explained later on. The last admissible status $p^{i'}$ is the last corresponding $p_t^i$ when $v_t^i$ is within $\pm 10\%$ of rated voltage. In order to make the control robust and stable, users are encouraged to trace back to their $p^{i'}$ if a regulation needs to be applied. Different from centralized control, there is no explicit set point given to users. If $p^{i'}$ is not applicable, user $i$ can keep the current working status and count 1 more on $c_i$. This gives user $i$ some time to wait for other users contribute to the necessary regulation. Whereas, if the situation remains when $c_i$ reaches to 2, user $i$ will do an experiment by adjusting its $p_t^i$ 2% towards the direction suggested by $\zeta_t^i$ and reset $c_i$ to zero. For the users who cannot adjust their working status by such a step, the closest working state will be chosen. Performing experiments is the least preferred operation as it will not be often when the game converges to a proper NE.

---

**Algorithm 3** The calculation of $\Delta p^i$

---

**Input:**

  Suggestion parameter, $\zeta_t^i$;
  initial plan , $\delta_t^i$
  Last admissible status, $p^{i'}$;
  Current working status, $p_t^i$;
  Trace-back count, $c_i$;
**Output:**

  Admissible adjustment, $\Delta p^i$;
  Trace-back count, $c_i$;
1: **if** $sgn(\delta_t^i) = sgn(\zeta_t^i)$ **then**
2:   $\Delta p^i = \delta_t^i, c_i = 0$;
3: **else if** $sgn(p^{i'} - p_t^i) = sgn(\zeta_t^i)$ **then**
4:   $\Delta p^i = p^{i'} - p_t^i, c_i = 0$;
5: **else if** $c_i <= 2$ **then**
6:   $\Delta p^i = 0, c_i = c_i + 1$;
7: **else**
8:   $\Delta p^i = 0.02 \cdot * p_t^i * sgn(\zeta_t^i), c_i = 0$
9: **end if**
10: Return $\Delta p^i, c_i$

---

As shown in Algorithm 2, $\Lambda_i$ is a vector that can affect the distribution of $s_i$, where $K = \frac{T}{\tau}$. $T$ is the timescale of $\Lambda_i$ while $\tau$ is the timescale of $[\lambda_m^i]$. In this paper, $T$ is 24 h and $\tau$ is 0.5 h, therefore $K = 48$. For each period $m$, there is a scalar pair $[\lambda_m^i]$, in which $\lambda_{1m}^i$ and $\lambda_{2m}^i$ are two thresholds for $s_i$ as illustrated in Algorithm 2. Namely, $\lambda_{1m}^i$ is the threshold between staying with the current working status and making a change towards the direction $\zeta_t^i$ suggests, while $\lambda_{2m}^i$ is the threshold between staying with the current working status and implementing change according to $\tilde{p}_{t+1}^i$. It benefits the whole LVDN most if user $i$ makes a change towards $\zeta_t^i$, while implementing change according to $\tilde{p}_{t+1}^i$ results in perfect user comfort. Scalar pair $[\lambda^i]$ indicates control policy of user $i$, as it configures the corresponding probability distribution of $\sigma_i \in M_i^\circ$. Given the situation that users have different behavioral characteristics during one day, different $[\lambda^i]$ is needed to seek for the NE in different games during different periods of one day. This is why $[\lambda_m^i]$ applies. In this paper, the optimal values of $[\lambda_m^i]^*$ are figured out by a trial and error learning approach illustrated by Algorithm 4, where sign *bino* stands for the Bernoulli trial.

In Algorithm 4, a variant of the so-called log-linear trial and error learning [43] is implemented. Note that $\zeta_t^i$ is given by a linear combination of two log models, and $[\lambda_m^i]$ is adjusted with probabilities in proportion to $J_i$ when $v_{t+\Delta t}^i$ approaches the median between $\gamma$ and technical boundaries ($\pm 10\%$ of $v^r$).

---

**Algorithm 4** Learning Phase

---

**Input:**

    Strategy, $s_i$;

    Local voltage , $v^i_{t+\Delta t}$;

    Corresponding scalar pair, $[\lambda^i_m]$;

    Boundary coefficient, $\gamma$;

    Suggestion parameter, $\zeta^i_t$;

    Adjust step, $\alpha = 0.02$;

**Output:**

    Corresponding scalar pair, $[\lambda^i_m]$;

1: calculate $\Delta v^i_{t+\Delta t}$ according to equation (8)
2: $\Xi = (0.99 - \gamma)/2$;
3: **if** $\zeta^i_t = 1$ **then**
4:     $\lambda^i_{1m} = \lambda^i_{1m} - \alpha$;
5:     $\lambda^i_{2m} = \lambda^i_{2m} - \alpha$;
6: **else if** $s_i=1$ **then**
7:     **if** $\Delta v^i_{t+\Delta t} < \gamma$ **then**
8:         $\lambda^i_{1m} = \lambda^i_{1m} + \alpha$;
9:     **else if** $\Delta v^i_{t+\Delta t} > 0.99$ **then**
10:         $[\lambda^i_m] = [\lambda^i_m]$; ## do nothing
11:     **else**
12:         $\lambda^i_{1m} = \lambda^i_{1m} - \alpha \cdot bino(4 \cdot (\Delta v^i_{t+\Delta t} - \Xi)) \cdot sgn(\Delta v^i_{t+\Delta t} - \Xi)$;
13:     **end if**
14: **else if** $s_i=0$ **then**
15:     **if** $\Delta v^i_{t+\Delta t} < \gamma$ **then**
16:         $\lambda^i_{2m} = \lambda^i_{2m} + \alpha$;
17:         **if** $\lambda^i_{2m} > \lambda^i_{1m}$ **then**
18:             $\lambda^i_{1m} = \lambda^i_{1m} + \alpha$;
19:         **end if**
20:     **else if** $\Delta v^i_{t+\Delta t} > 0.1$ **then**
21:         $\lambda^i_{1m} = \lambda^i_{1m} - \alpha$;
22:         **if** $\lambda^i_{1m} < \lambda^i_{2m}$ **then**
23:             $\lambda^i_{2m} = \lambda^i_{2m} - \alpha$;
24:         **end if**
25:     **else**
26:         $\lambda^i_{2m} = \lambda^i_{2m} - \alpha \cdot bino(4 \cdot (\Delta v^i_{t+\Delta t} - \Xi)) \cdot sgn(\Delta v^i_{t+\Delta t} - \Xi)$;
27:     **end if**
28: **else if** $s_i=-1$ **then**
29:     **if** $\Delta v^i_{t+\Delta t} < \gamma$ **then**
30:         $\lambda^i_{1m} = \lambda^i_{1m} + \alpha$;
31:         $\lambda^i_{2m} = \lambda^i_{2m} + \alpha$;
32:     **else if** $\Delta v^i_{t+\Delta t} > 0.1$ **then**
33:         $\lambda^i_{2m} = \lambda^i_{2m} - \alpha$;
34:     **else**
35:         $b = bino(4 \cdot (\Delta v^i_{t+\Delta t} - \Xi)) \cdot sgn(\Delta v^i_{t+\Delta t} - \Xi)$;
36:         $\lambda^i_{1m} = \lambda^i_{1m} - \alpha \cdot b$;
37:         $\lambda^i_{2m} = \lambda^i_{2m} - \alpha \cdot b$;
38:     **end if**
39: **end if**
40: **if** $\lambda^i_{1m} > 1 - \alpha$ **then**
41:     $\lambda^i_{1m} = 1 - \alpha$;
42:     **if** $\lambda^i_{1m} \leq \lambda^i_{2m}$ **then**
43:         $\lambda^i_{2m} = \lambda^i_{1m} - \alpha$;
44:     **end if**
45: **else if** $\lambda^i_{2m} < \alpha$ **then**
46:     $\lambda^i_{2m} = \alpha$;
47:     **if** $\lambda^i_{2m} \geq \lambda^i_{1m}$ **then**
48:         $\lambda^i_{1m} = \lambda^i_{2m} + \alpha$;
49:     **end if**
50: **end if**
51: Return $[\lambda^i_m]$

---

### 3.3. Remarks

The proposed "suggest-convince" mechanism is essentially a simulation of the negotiation process in games. User $i$ employs public information $g_t$ and its local information $v_t^i$, to generate $\zeta_t^i$, which indicates the power changes preferred by the circumstance. Then compare $\zeta_t^i$ with the three zones divided by corresponding stubborn scalar pair $[\lambda_m^i] \in \Lambda_i$, to figure out $s_i \in \mathbb{S}_i$. Eventually, one needs to review $v_{t+\Delta t}^i$ to see whether there is potential to improve $E(J_i)$, then adjust the corresponding $[\lambda_m^i]$, in order to approach the best mixed strategy $\sigma_i^* \in M_i^\circ$ during the given period $m$.

Within a given period $m$, assume user $i$ has more moderate profile that leads to milder $\delta_t^i$ than user $j$, $i \neq j$, or the neighboring users of $i$ are more supportive to grid regulation, $v_{t+\Delta t}^i$ will be more likely stay within or closer to $(1 \pm \gamma)v^r$ than $v_{t+\Delta t}^j$, which results in $\left(s_i^*|i\right) \succsim \left(s_j^*|j\right)$ or $\left(\sigma_i^*|i\right) \succsim \left(\sigma_j^*|j\right)$ depending on the network configuration. According to Proposition 2 and 3, this relationship is transitive and asymmetric, it spreads through the whole LVDN via the coupling among the users. It is important to point out that this relationship is not necessarily uniform in LVDN, as the coupling among users depend on the network topology, it is possible to have several player compatible relations in a given LVDN. One of the objectives of Algorithm 4 is discovering such a relation, encouraging users who have the upper hand to maximize their probabilities on strategies $s_i = 1$ and $s_i = 0$ within restrictions. Meanwhile, the boundary check on $[\lambda_m^i]$ in Algorithm 4 (lines 40–49) guarantees the existence of $\Pi_i^\epsilon$, therefore $\epsilon$-*equilibrium* exists. Consequently, as stated in (1), user $i$ has a finite strategy set $\mathbb{S}_i = \{-1, 0, 1\}$. Therefore, this is a finite strategic-game and PCE exists according to Theorem 2.

Ref. [43] suggests that in an interdependent N-person game with a finite strategy set, if all players use log-linear trial and error learning, and that the acceptance probabilities are fairly large relative to the probability of conducting an experiment, then its stochastic stable state will be either a pure NE or mixed strategy that maximizes $\sum_{i=1}^{N} U_i$ if pure NE does not exist. For user $i$ in this paper, the "acceptance probabilities" are the probabilities of $v_{t+\Delta t}^i$ will stay within or closer to $(1 \pm \gamma)v^r$, and the "probability of conducting an experiment" is its tremble profile $\epsilon$. The conditions are satisfied. Therefore, the stochastically stable state of user $i$ will be PCE. Besides, Definition 2 essentially supposes that user $i$ considers the set of all mixed correlated strategies of other users $\sigma_{-i} \in M^\circ(\mathbb{S}_{-i})$. If the players can learn some prior knowledge about their counterparts' player compatibility, user $i$ might be able to deduce that the counterparts will only play subset $\hat{M}_{-i} \in M^\circ(\mathbb{S}_i)$. This prior knowledge can be obtained by Algorithm 4 in this paper, so that the convergence of $\sigma_i$ can be expected.

Additionally, some parameters are tunable. $\gamma$ is set to 0.5 in this paper, which means the control is trying to make voltages converge to $\pm 5\%$ of $v^r$ instead of itself. This makes more sense to the proposed approach as it allows users to take advantage of more flexibility to improve comfort, but still with certain margin. It is necessary to point out that larger $\gamma$ does not lead to more available flexibility, the optimal $\gamma$ still needs further study. $\tau$, as the timescale of $[\lambda_m^i]$, affects the converge speed and quality. Although the behavior of users in LVDNs is changing, it is relatively stable hourly and has daily characteristics. This is why $\tau$ is 0.5 h and $T$ is 24 h in the paper. It allows user $i$ to employ $[\lambda_m^i]$ and $\Lambda_i$ to learn the hourly statistic characteristics and fit daily dynamic patterns respectively from their counterparts.

## 4. Case Study

### 4.1. Grid Topology

The schematic diagram is illustrated in Figure 1. It is a three-phase 230/400-V reference grid based on the topology of a real semiurban feeder in the region of Flanders, Belgium [44]. To make the network more multifarious and ill-designed, new users such as residential wind turbines and small PV farms are added, increasing the number of nodes from 62 to 103. As listed in Table 1, the impedance values are calculated according to the Belgian standard for underground distribution cables with an assumed operating temperature of 45 °C. All of the main feeder cables are of type EAXVB 1 kV

$4 \times 150$ mm$^2$. A 250 kVA 10/0.4 kV transformer is assumed with an impedance of 0.013 + 0.038j pu. From feeders to each individual user,/hl a cable EXVB 1kV $4 \times 16$ mm$^2$ is used with a length of 15 m. To simplify, the three-phase system is assumed to be symmetrical, then the analysis can focus on a single phase. The $v^r$ of all the users is 230 V.

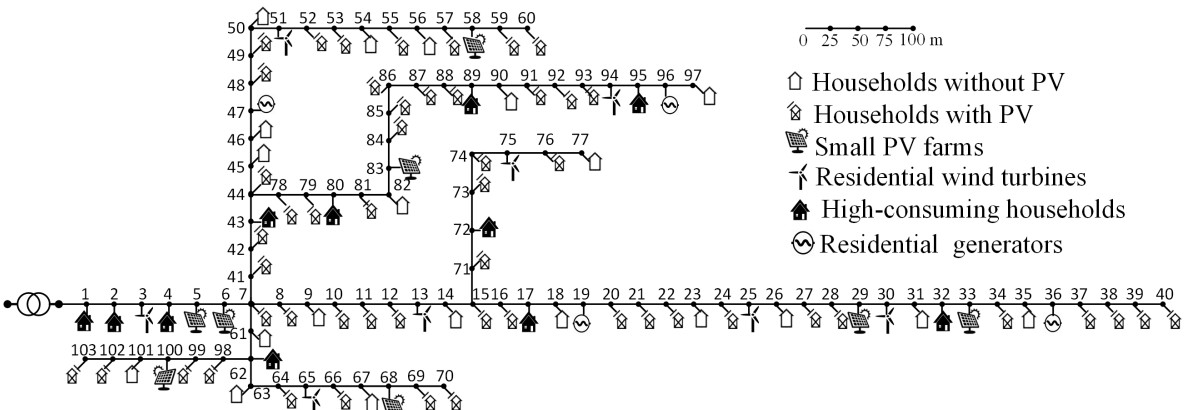

**Figure 1.** The topology of the test network. Cable lengths are drawn to scale.

**Table 1.** Cable parameters.

| Cable Type | Impedance at 45 °C |
|---|---|
| EAXVB 1kV $4 \times 150$ mm$^2$ | 0.227 + 0.078j Ω/km |
| EXVB 1kV $4 \times 16$ mm$^2$ | 1.265 + 0.083j Ω/km |

### 4.2. Profiles and Conditions

A model of domestic electricity use is used to generate the high-resolution household consumption profile, which is based upon a combination of patterns of active occupancy [45]. Lighting and appliances, occupant's behavior, month of the year, and weekday or weekend have been taken in account for the profile generation. Household profiles in this paper are assumed on a weekend day of June, whose source database is based on a realistic measurement of 22 domestic dwellings around the town of Loughborough in the East Midlands, UK. It is assumed that normal households randomly have one to four family members, and four to six family members are assumed in high-consuming households. Profiles of PV panels are taken from a 368 kWp PV system on the rooftop of EnergyVille in Genk, Belgium [46]. The system has 24 strings of PV panels, and each string can be recorded individually. The profiles are scaled, randomly selected and combined from the raw data in the week of 6th–12th June, 2016. As all the PV strings are located close to each other, the data has a good correlation, which is suitable to be used as a profile in the LVDN. Regarding residential wind turbines, their profiles are obtained from Elia, a Belgian transmission system operator. The data was measured from Belgian onshore wind turbines on the 6th and 7th of June, 2016, the same time period as the profiles of PV and households. Then it is scaled and assigned randomly to the residential wind turbines in the test network. Four small residential generators are connected to the test network to represent other distributed generation users, such as CHP generators. Their profiles are generated randomly taking the occupants' behavior into account. The operation ranges of users are given in Table 2.

**Table 2.** Operating ranges of users (negative values mean production, positive values consumption of electricity).

| User Type | Operating Range (kW) |
|---|---|
| Houses without PV | 0~13 |
| Houses with PV | −2~13 |
| Small PV farms | −11.5~0 |
| Residential wind turbines | −20~0 |
| High-consuming unit | 2~17 |
| Residential generators | −7~0 |

In LVDNs, due to the high R/X ratio, voltage is more related to active power distribution [37]. Moreover, users, especially households in LVDN do not have any devices of which the reactive power output can be controlled. To simplify the case, assume that users only consume active power or the users' power factor is at least 0.9. This does not mean that the reactive power is ignored in the control, as users consider the actual voltage changes. The associated influence caused by other factors, such as reactive power changes due to the control, have already resulted in the actual voltage changes. For instance, although the incentive of a user to turn on the washing machine is active power, the associated reactive power is changed as well.

*4.3. Simulation Setup*

The algorithm runs once per minute, which means 30 times during each $m$. The initial $[\lambda_m^i]$ is set to $[0.5, 0.2]$ uniformly. The power flow simulation is implemented in MATLAB, based on the backward-forward sweep method [47]. Although the control is simulated step by step, its effect is processed as continuous. Namely, to make the simulation more realistic, user behaviors are extracted from the original profile. In every simulation step, we assume that the users will restore half of the regulations (if any) on them from the previous step, to simulate the decaying continuous effect from the control implemented before. Therefore, the reference profile in each step is the combination of corresponding user behaviors and the decayed previous status, instead of just obtaining status from the original profile rigidly. Meanwhile, to simulate the daily characters from different days, we use the same original profiles for all the days, nevertheless, small random generated variations are added to $p^i$ to make the operation conditions in the simulation not remain exactly the same among different days, which makes it more realistic.

*4.4. Simulation Results and Remarks*

To present a basic view on the convergence process, the average voltage from all the users are shown in Figure 2a. The test profile has three typical scenarios: from 0–8 h, the original profile is moderate, the users gradually adapt themselves to form an equilibrium situation; from 10–16 h, the system suffers from severe overvoltage due to the generation by PV modules, users finally reach an equilibrium that employs the 5% margin to avoid bothering the users as much as possible. From 18–24 h, the mild voltage variation does not motivate most of the users, so that they increase their corresponding $[\lambda_m^i]$ to guarantee their comfort. To study this in numbers, the discord rate $D_m$ is defined as

$$D_m = \sum_{t=0}^{\tau} \sum_{i=1}^{N} \Psi(s_i(g_t, v_t^i), \delta_t^i), \tag{9}$$

where $\Psi$ is the operation that counts the difference between the actual changes $p_{t+\Delta t}^i - p_t^i$ and $\delta_t^i$: if $\delta_t^i$ is eventually implemented, it counts 0, if does not, it counts 0.5, and it counts 1 if the actual adjustment opposites to what $\delta_t^i$ demands. We do not use $J_i$ here as $J_i$ is a combination of user comfort and technical demands, which will be shown later on. The hourly $D_m$ is illustrated in Figure 2b. From day 1 to day 4 one can figure out that along with the form of PCE, the discord rate is decreasing, which

suggests better user comfort and smoother control. A centralized control that can derive the optimal adjustment for each participating user is implemented [48,49] as the benchmark. The controller solves an optimization problem by the full non-linear network model (Alternating Current Optimal Power Flow, ACOPF). It employs two-way communication; each participating user sends a range to the central controller, within which it can adjust its power consumption or generation. Besides, assumptions of perfect information, instant computation, communication, and implementation are given to the control, to obtain the theoretical best result to be compared with. The objective is minimizing $(p^i_{t+\Delta t} - \tilde{p}^i_{t+1})^2$, with the boundary condition of $0.95 \cdot v^r \leq v^i_t \leq 1.05 \cdot v^r$. The general average voltage is shown in Figure 3a, while the comparison of the discord rate is illustrated in Figure 3b. The statistical results of general voltage and nodal voltages are shown in Table 3.

**Table 3.** Performance statistics of individual users.

| Item | General Voltage Deviation Range (%) | $10\% < \lvert \Delta v^i_t \rvert$ | $\lvert \Delta v^i_t \rvert \leq 10\%$ |
|------|-----|-----|-----|
| No control | $-6.34 \sim +18.07$ | 28.08% | 71.92% |
| ACOPF | $-5.39 \sim +5.47$ | 0% | 100% |
| Day 1 | $-4.85 \sim +10.03$ | 0.26% | 99.74% |
| Day 2 | $-6.14 \sim +9.55$ | 0.19% | 99.81% |
| Day 3 | $-6.00 \sim +8.93$ | 0.12% | 99.88% |
| Day 4 | $-6.01 \sim +8.66$ | 0.12% | 99.88% |

From Figure 3 it can be concluded that the proposed approach has achieved a good approximation to a centralized optimization solution. Meanwhile, the proposed approach comes with a relatively lower discord rate. This does not mean that the proposed approach is better than a centralized approach in all the aspects, as the centralized approach has a rigid boundary, while the proposed approach statistically converges to the boundary. This can be observed in Figure 4, which shows all the 103 nodal voltages of the proposed approach and centralized approach. It is clear that the proposed approach does not have an absolute hard boundary—it stochastically allows users to use the preserved margin to some extent, while the conventional ACOPF uses the absolute hard boundary to guarantee the preserved margin, whereas it comes with a higher discord rate. Besides, few overvoltages of nodes can be observed from Figure 4a, this is caused by drastic power changes. As it is model-free, the proposed approach is not able to have an accurate prediction on voltage changes, when the general condition is mild at time $t$, it is very likely to result in corresponding mild $\zeta^i_t$, which gives users a higher probability on implementing $\tilde{p}^i_{t+1}$ to maximize the user's comfort. If $\tilde{p}^i_{t+1}$ indicates a dramatic change compared to $p^i_t$, the slight overvoltage might happen. Nevertheless, it is fixed immediately in the next control period.

To show how the PCE is formed and the evolution of the individual user, Figure 5 illustrates the changing of $[\lambda^i_m]$ in all the four days continuously, user $i = 36$ is selected randomly as an instance. Combining the figures of $\lambda^i_{1m}$ and $\lambda^i_{2m}$, the evolution of the two thresholds of different $s_i$ in $\mathbb{S}_i$ is changing. During some $m$, due to flexible neighbors or mild user profile, user $i = 36$ remains the upper hand all the time, such that both of the two scalars in its $[\lambda^i_m]$ reach their maximum, to take advantage of the deregulation as much as possible. Whereas for some $m$, user $i = 36$ increases its $[\lambda^i_m]$ at the beginning to feel out the other users, then it has to restrain its scalars as it does not have actual superiority compared to others.

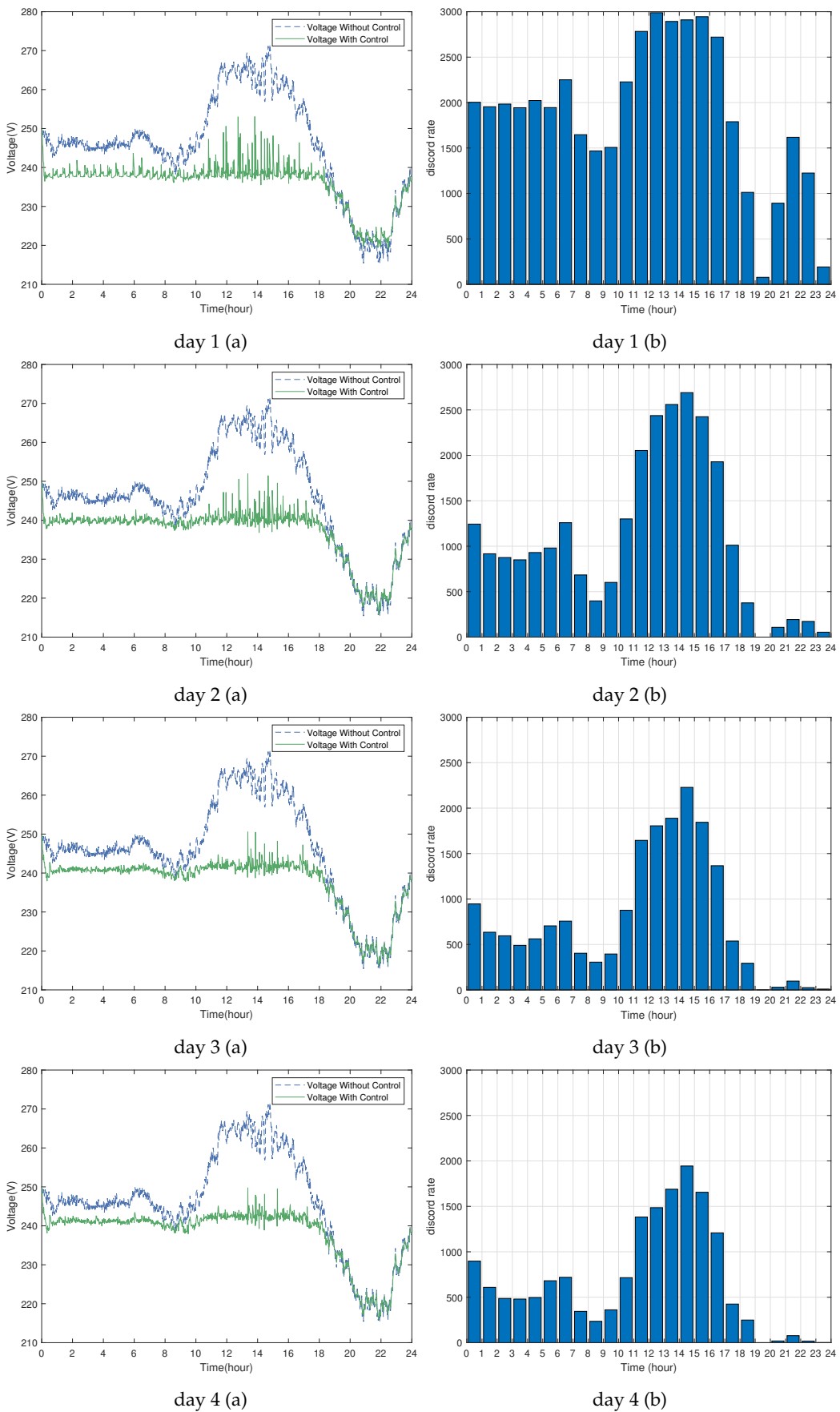

**Figure 2.** General voltage (**a**) and discord rate (**b**) from day 1 to day 4.

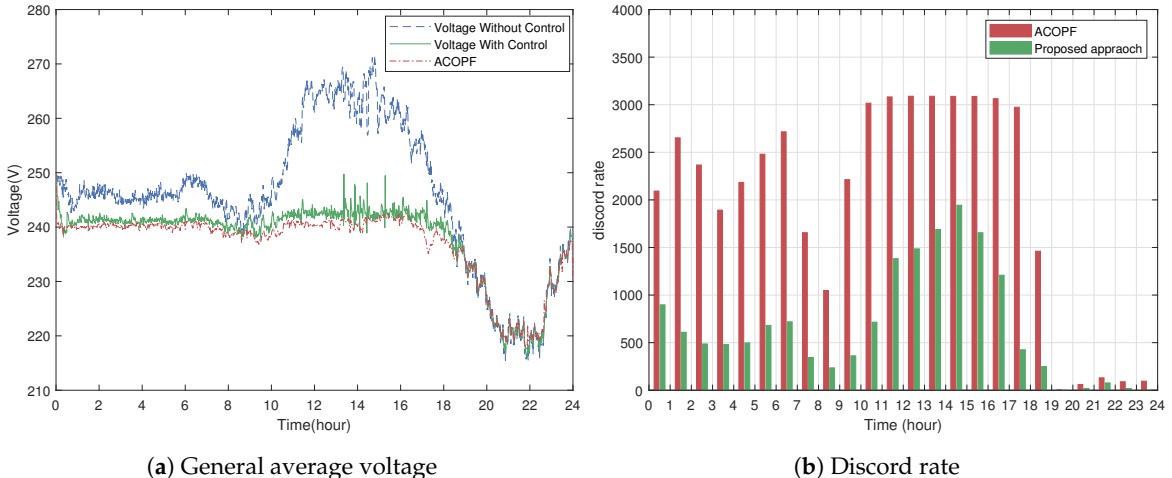

(**a**) General average voltage                              (**b**) Discord rate

**Figure 3.** Comparison between proposed approach (day 4) and centralized approach.

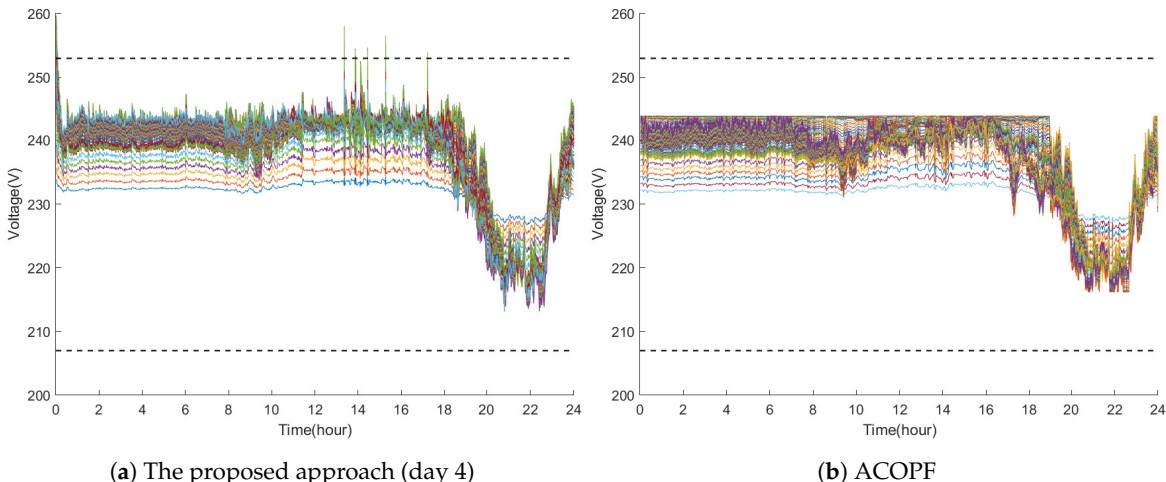

(**a**) The proposed approach (day 4)                              (**b**) ACOPF

**Figure 4.** Nodal voltages of the proposed approach (day 4) and centralized approach.

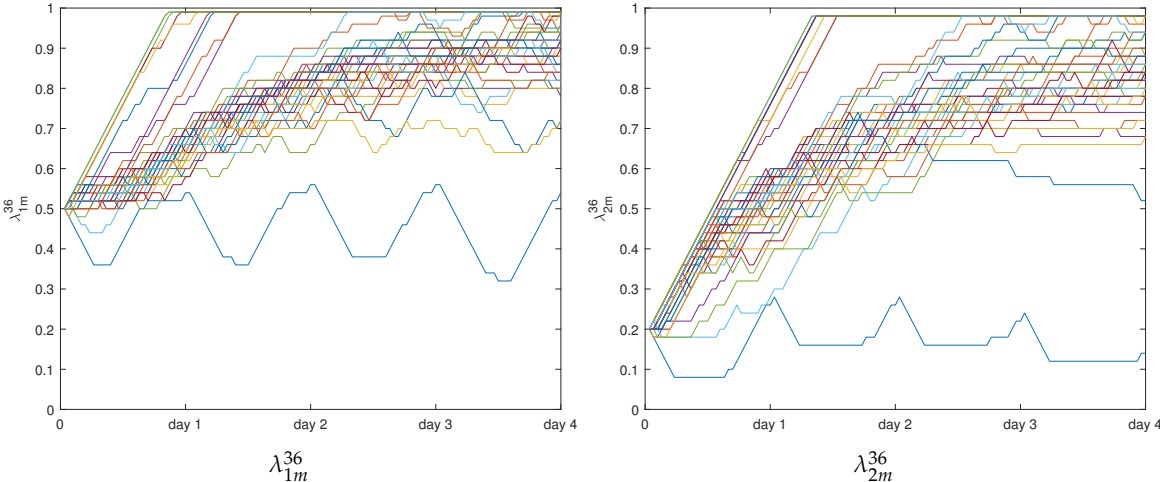

$\lambda_{1m}^{36}$                                                $\lambda_{2m}^{36}$

**Figure 5.** The evolution of $[\lambda_m^i]$ in four days, $i = 36$.

The global convergence can be observed from Figure 6. The gross discord rate is given by $\sum_{m=1}^{24} D_m$ everyday, while nodal voltage deviations are exactly in the manner of Table 3 but for 7 days. It can be observed that, with the current configuration, from the fourth day, the whole system reaches a statistically steady state, which is supposed to be PCE according to the derivations in Sections 2

and 3. Users achieved their approximate local optimum in the non-cooperative game. Four days are not a short time, nevertheless, although coming with high gross discord rate, the system gets well controlled from the first day, then the equilibrium is gradually formed via trial and error learning process. As long as the equilibrium is formed, it is robust unless the whole LVDN gets completely changed, which makes the LVDN hot-plugging and flexible. These features are in line with the concept of ADN as well.

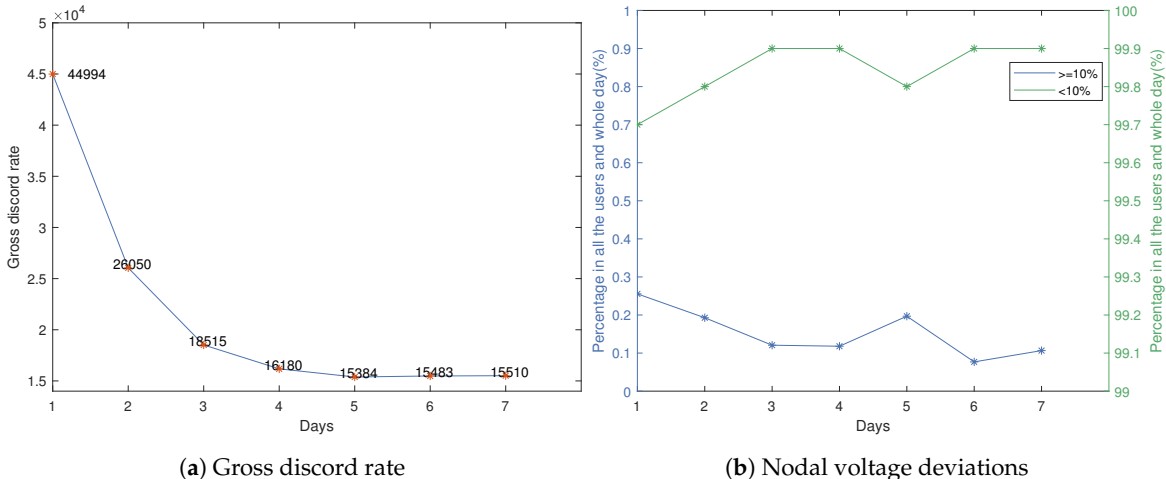

(**a**) Gross discord rate      (**b**) Nodal voltage deviations

**Figure 6.** The evolution of gross discord rate and nodal voltage deviations.

## 5. Conclusions

A model-free control approach with distributed decision-making architecture is proposed in this paper. With statistic and game theories, it achieves good approximation to local optimum among individual users in the LVDN. Compared to conventional approaches applied in the LVDN, the proposed approach is able to achieve active control with a low communication burden and computational resources. Users and broadcasters are double-blind to each other, which allows users to enter or quit the network freely (i.e., hot-plugging). Moreover, there is only anonymous general information delivered through the whole network, which addresses the concern on privacy. These make the proposed approach in accordance with the developing trend of privacy protection and decentralization in the LVDN.

Although not many, there are some existing works that proposed voltage control by game theoretical algorithms. For instance, Zhou et al. [50] employs Volt/VAr control dynamics with nonlinear power flow model to do a voltage control game. Compared to the proposed approach, as [50] works with an explicit model to indicate the influence on the voltage that users can exert by changing their consumption, a more accurate result could be expected if the model is well designed. Nevertheless, if the gradient of its piecewise linear volt/var control curve is too large, the algorithm may have a convergence problem, which is not a problem for the proposed approach as its convergence is guaranteed by the log-linear trial and error learning process and the approach itself is model-free. Nassaj and Shahrtash [51] employs the Shapley–Shubik index to implement dynamic voltage control in the distribution network. Normally this approach needs to figure out the Shapley values by communication before starting the game; Nassaj and Shahrtash [51] calculates the Shapley–Shubik indices, and then distributes them to implement the control, which is essentially not a completely distributed control.

Theoretically, the proposed approach can be applied in other scenarios as well, as long as the control can be formed as a game that meets two prerequisites. Firstly, it should be a potential game, which mainly means the congestion game in the power system. Secondly, there should be a hierarchical relationship in the aspect of priority among interdependent controlled agents, no matter whether this relationship is connatural or given on purpose. For instance, the control of OLTC and

elastic loads in [52], smart EV charging in [53], the demand side management of large populations of thermostatically controlled loads [54], and frequency control with energy storage systems in the distribution network [55].

Although the proposed approach is able to minimize $J_i$ by reaching PCE, it does not guarantee global optimum, as the NE is an equilibrium among users, where every user $i$ is close to its local optimum with limitations. The global optimum could be able to be achieved via peer to peer communication, with sophisticated algorithms and configuration according to the Fundamental Theorems of Welfare Economics [56]. It has the potential to promote Pareto improvement on the achieved NE, and eventually reach a statistically steady point on the Pareto Frontier. This will be the focus of future research as seeking the Pareto Frontier is one of the classic ways to solve multi-objective optimization problems.

**Author Contributions:** Conceptualization, B.W. and G.D.; methodology, B.W.; software, B.W.; validation, B.W.; formal analysis, B.W.; data curation, B.W.; writing–original draft preparation, B.W.; writing–review and editing, B.W. and G.D.; visualization, B.W.; supervision, G.D.; project administration, G.D.; funding acquisition, G.D. All authors have read and agree to the published version of the manuscript.

**Funding:** This research was funded by CSC, VLAIO Flux50 ICON HBC.2018.0527 ROLECS "Roll-out of Local Energy Communities", and KU Leuven BOF/IOF C24/16/018 "Energy Storage as a Disruptive Technology in the Energy System of the Future".

**Conflicts of Interest:** The authors declare no conflict of interest.

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
