# Peer review of "Distributed Optimization in Low Voltage Distribution Networks via Broadcast Signals ††"

_energies, doi:10.3390/en13010043_

Round 1
Reviewer 1 Report
Line 2: The authors need to specify what two things LVDN interface with. Is it the interface between small distributed energy resources and the distribution grid or subtransmission system? For example, someone might say a salesman is the interface between the customer and the product. You cannot say the salesman is the interface between the product. The sentence is an incomplete thought. Line 3: List some of the challenges with modeling you are referring to. Line 3-4: The sentence "Additionally, data acquisition is hardly feasible due to privacy concerns" seems out of place and dismissive. It is out of place because classical centralized control does not assume to collect data that would cause privacy concerns; thus the two sentience's ideas seem disconnected. Secondly, while I agree privacy issues are a concern, that doesn't mean this problem won't be solved in the future. There are many research efforts aimed at enhancing privacy and security while offering incentives to encourage users to share their data. If the authors wish to keep this sentence, they should move it after the sentence "these impose severe demands..." and then add to the sentence a list of control schemes that require user data with privacy concerns. This will make the paper read much better and the arguments appear stronger. Line 11: This sentence "A variant of log linear trial and error learning process is applied in a novel "suggest-convince" mechanism as the implementation." is not a complete through. The authors must finish the sentence by answering "as the implementation of what?" Is it the implementation of LVDN control? Line 13: Are you refereeing to load profiles? Please state what kind of profile is being referring to. Line 15: The statement "in order to have a closer look to the proposed approach." is vague and uninformative. Typically, when comparing a new system to a baseline case, one is trying to understand how the new systems is better or worse than the baseline. Please state why you are comparing centralized control to your approach by talking about what you hoped to learn by doing so. Line 21: Change to "The rising penetration..." Line 23: Consider "...as LVDN is expected to facilitate DERs penetration." Line 25-27: This sentence is a little confusing "On the other hand, consumer privacy becomes an important concern when controlling users with the deployment and adoption of smart grid technologies, which set severe barriers to the data collection from users [1]." Consider rewording to "On the other hand, consumer privacy becomes an important concern when utilities aim to control residential loads with the deployment of smart grid technologies [1]." Line 27-29: Please list out some of these control approaches that are being referred to and also provide several citations for them. Line 31: This is an incomplete sentence and thought, please reword and state what the goal referred to is. Line 33-34: This paragraph is not organized well because the sentences don't connect well. Consider rewording the sentence "Nevertheless..." to something like "The previous issue of load sharing with droop control can be solved by..." or something like that. Line 36 This makes no sense "...which varies alone network configuration." please clarify. Line 39: Achieving does not seem like the correct word here. Please fix this or reword the sentence so that the meaning is clear. Line 40: Change "users" 'to "they" Line 42: Change "appreciated" to "needed" Line 42-44: Is the paper trying to say that user's data will only be used locally and therefor mitigate privacy concerns? If yes, it is currently worded in a very confusing manner and must be corrected. Considered expanding a bit on the idea instead of trying to say too much with too few sentences. Line 46: Is this referring to system size? If so, state that. line 50: Change to ""Decomposition [11] and decentralized..." Line 61: Change "applications :" to "applications:" Line 62: replace ";" with "and" Line 64: Chang to "...in the control..." Line 99-100: What is meant by rated power in the following statement: "The users are not necessarily homogeneous but share a certain degree of similarity in terms of rated power."? Is it saying that they all consume or provide power on a similar order of magnitude? Please clarify. Line 103: add the word "which" to "...in power balance, which leads to..." In the paragraph after line 112 (why are the lines not numbered on that paragraph?) second sentence, change to: "...reserve is not sufficient, this leads to a voltage problem..." In the paragraph after line 112 third sentence, change to: "...is supported by a backbone..." Line 150: change to "...can be regarded as a flexibility allocation game." Line 156: delete the word "exists" line 158: change to "...converge to a NE within an acceptable time..." Line 160-161: change to "...In other words,..." Line 162: Did the authors mean "In practice,..." or "Practically speaking,..." Line 232: Could you state the voltage regulation standard that you used to choose 5% of V^r? Line 277: Change to "...incentive of a user..." Line 290: delete "does" and add "the" between "... exactly same..." Line 301: Change "two-ways" to "two-way" Line 325-326: Change to "...user I=36 retains the upper hand at all times." Line 328-329: This part of the sentence isn't clear to me "...at the beginning as all the other are sounding out as well, then it has to restrain its scalars since it does not have actual superiority comparing to others." please clarify what the is referred to by "sounding out" line 343: Change to "...with a low communication..." Line 344-345: Change "Besides, as users and broadcaster are double blind to each other, it allows users enter/quit the network freely, namely, hot-plugging." to "Users and broadcaster are double blind to each other, which allows users to enter or quit the network freely (i.e. hot-plugging)." or "Because users and broadcaster are double blind to each other, users are enabled to enter or quit the network freely (i.e. hot-plugging)." Line 346: delete "...public and..." and "...well." Very interesting paper. The authors should include a short discussion on how their algorithm's results compare to previously used game theory algorithms. Was it better or worse at controlling system voltage? In what ways was it better or worse? Lastly, could the authors comment on the stability of the simulation from multiple runs? Does it always reach PCE every time? Do some simulations have more over voltage issues than others? How well does the data presented represent this algorithm overall?
Reviewer 2 Report
Dear Authors,
Application of game theory for control of LVDN networks is an interesting idea.
On the flip side the proposed algorithm appears to be very complicated for a simple voltage control in the distribution network.
It has been mentioned that the advantage of the proposed approach is low communication burden and needs less computational resources. This might be true but it is no longer true with the present day computational capabilities.
Kindly provide more applications of the proposed approach in addition to voltage control. If possible kindly add additional results.
Reviewer 3 Report
- present with more details the other researchers work in the introduction;
- specify the source of the equations;
- what is the installed power of the PV, wind turbines and CHP?
- how does the installed power of the PV, wind turbines and CHP affect the results?
- expand the conclusion section by referring to the simulation results;
- at least half of the last page should be covered with text.
Round 2
Reviewer 2 Report
Dear Authors,
Thanks for responding to my queries. Kindly review the paper thoroughly before submitting the final version.